# Effect and Tolerability of a Nutritional Supplement Based on a Synergistic Combination of β-Glucans and Selenium- and Zinc-Enriched *Saccharomyces cerevisiae* (ABB C1^®^) in Volunteers Receiving the Influenza or the COVID-19 Vaccine: A Randomized, Double-Blind, Placebo-Controlled Study

**DOI:** 10.3390/nu13124347

**Published:** 2021-12-02

**Authors:** Julián Andrés Mateus Rodriguez, Mónica Bifano, Elvira Roca Goma, Carlos Méndez Plasencia, Anna Olivé Torralba, Mercè Santó Font, Pedro Roy Millán

**Affiliations:** 1Hospital Mare de Déu de la Mercè, Hermanas Hospitalarias, 08042 Barcelona, Spain; mbifano.merced@hospitalarias.es (M.B.); cmendez.merced@hospitalarias.es (C.M.P.); aolive.merced@hospitalarias.es (A.O.T.); msanto.merced@hospitalarias.es (M.S.F.); proym.merced@hospitalarias.es (P.R.M.); 2Clinica Nostra Senyora del Remei, 08024 Barcelona, Spain; 3CBC Isabel Roig, 08030 Barcelona, Spain; 4Unitat Polivalent Barcelona Nord, Hermanas Hospitalarias, 08035 Barcelona, Spain; eroca.merced@hospitalarias.es

**Keywords:** trained immunity, influenza vaccine, COVID-19 vaccine, β-1,3/1,6-glucan complex, selenium, zinc, *Saccharomyces cerevisiae*, nutritional supplementation

## Abstract

A single-center, randomized, double-blind, placebo-controlled study was conducted in 72 volunteers who received a synergistic combination of yeast-based ingredients with a unique β-1,3/1,6-glucan complex and a consortium of heat-treated probiotic *Saccharomyces cerevisiae* rich in selenium and zinc (ABB C1^®^) or placebo on the next day after getting vaccinated against influenza (Chiromas^®^) (*n* = 34) or the COVID-19 (Comirnaty^®^) (*n* = 38). The duration of treatment was 30 and 35 days for the influenza and COVID-19 vaccine groups, respectively. Mean levels of CD4+T cells increased from 910.7 at baseline to 1000.2 cells/µL after the second dose of the COVID-19 vaccine in the ABB C1^®^ group, whereas there was a decrease from 1055.1 to 929.8 cells/µL in the placebo group. Changes of CD3+T and CD8+T lymphocytes showed a similar trend. In the COVID-19 cohort, the increases in both IgG and IgM were higher in the ABB C1^®^ supplement than in the placebo group. Serum levels of selenium and zinc showed a higher increase in subjects treated with the active product than in those receiving placebo. No serious adverse events related to ABB C1^®^ or tolerance issues were reported. The study findings validate the capacity of the ABB C1^®^ product to stimulate trained immunity.

## 1. Introduction

There is consistent evidence that healthcare workers on the frontline, older adults, and particularly elderly people with comorbidities have been at higher risk and poorer outcome when infected by the new Severe Acute Respiratory Syndrome Coronavirus 2 (SARS-CoV-2) [1,2]. Reduced effectiveness of the immune system associated with advanced age (immunosenescence) has shown to be a contributory factor in the development of severe COVID-19 [3]. In the current situation of the ongoing pandemic caused by SARS-CoV-2 and the probable coincidence with the annual influenza epidemic, assessment of the role that the influenza vaccine may play in COVID-19 disease is of utmost importance [4].

Different studies have shown a significant inverse association between influenza vaccination and severity of COVID-19 in terms of hospitalized patients, ICU admissions, and deaths attributable to the virus [5,6,7], raising the possibility that the influenza vaccine may activate the immune system sufficiently to prevent infection (by any virus) in a window after vaccination [4]. It was suggested that this could occur if the vaccine is able to sufficiently stimulate innate immunity against respiratory viruses, including SARS-CoV-2. On the other hand, innate immunity induced by certain life vaccines, such as the tuberculosis vaccine bacillus Calmette–Guérin (BCG) [8] or bioactive compounds, such as yeast β-1,3/1,6-glucans [9], can adapt and respond more efficiently to a second exposure through epigenetic, transcriptional, and functional reprogramming, namely, “trained immunity” [10,11]. In this respect, the activation of macrophages, dendritic cells, natural killer cells, and neutrophils due to trained immunity induced by β-glucan could possibly aid immune responses against SARS-CoV-2 and could help to prevent a severe clinical course [12].

Nutritional supplements oriented to improve immune defense mechanisms for an effective antiviral immune response have been an increasing focus of interest, particularly in the face of immunosenescence and to compensate for specific deficiencies of micronutrients among elderly subjects. Selenium is an essential trace element for a well-balanced immune response [13] and plays an important role in antioxidant defense [14], redox signaling pathways [15], and redox homeostasis [16]. An association between selenium deficiency and higher viral pathogenicity and an increased mortality risk from COVID-19 was shown [17,18,19]. In a similar way, zinc is a critical trace mineral for antiviral immunity [20], and low zinc levels at admission have been reported to be associated with poor clinical outcomes in SARS-CoV-2 infection [21]. In the presence of an immune function challenge, such as influenza vaccine, dietary interventions with selenium alone [22] or combined with zinc [23] have reported to increase humoral response. Accordingly, an adequate supply of selenium and zinc may be a useful intervention to mitigate the course of COVID-19 [24] and optimize vaccine efficacy [25].

*Saccharomyces cerevisiae* yeasts enriched with selenium or zinc improve the bioavailability and immunostimulatory properties of these organically bound trace minerals [26,27], and, in addition to of their carrier function, *S. cerevisiae* exerts beneficial effects on the gut microbiome by modulating immunity and binding and neutralizing pathogenic enteric bacteria and bacterial toxins [28]. In this respect, gut microbiota dysbiosis has been related to the immune response and persistent symptoms in COVID-19 patients [29,30,31].

On the other hand, in murine models of cancer, the addition of selenium improves the immunomodulative effects of β-glucan, with suppression of tumor growth, probably via stimulation of immunity [32,33].

The present study was conducted to assess the effect and safety of a nutritional supplement based on a synergistic combination of yeasts’ β-glucans with selenium and zinc-enriched *S. cerevisiae* (ABB C1^®^) [32] in two populations of elderly volunteers receiving the influenza vaccine and the COVID-19 vaccine. The aim of the study was to determine whether nutritional supplementation could improve the immune response to these vaccines and the micronutrient status of the participants. The high tolerance, safety, and immediate availability of ABB C1^®^ make this product an ideal candidate for dietary management in all types of subjects receiving the influenza or COVID-19 vaccines including geriatric and immunocompromised populations.

## 2. Materials and Methods

### 2.1. Study Design

This was a single-center, randomized, double-blind, placebo-controlled trial that was conducted at Hospital Mare de Déu de la Mercè and Unitat Polivalent Barcelona Nord belonging to FIDMAG Hermanas Hospitalarias Research Foundation (Barcelona, Spain). The primary objective of the study was to assess changes in CD4+T lymphocytes after dietary supplementation with ABB C1^®^ in two cohorts of people receiving the influenza or the COVID-19 vaccine. Besides changes in CD4+T cells, biomarkers of immunity included blood levels of CD3+T and CD8+T cells and levels of specific antibodies (IgM and IgG) against influenza or SARS-CoV-2 at the end of the supplementation period as compared with baseline. The secondary objective was the measurement of serum levels of selenium and zinc.

The trial was performed between October 2020 and June 2021. The study protocol was approved by the Clinical Research Ethics Committee of FIDMAG Hermanas Hospitalarias Research Foundation (code PR-2020-13) and was registered in the ClinicalTrials.gov (NCT04798677). The study was conducted in accordance with the Declaration of Helsinki. All participants gave written, informed consent. Participants’ records and information were anonymized.

### 2.2. Participants’ Selection and Randomization

For the influenza vaccination cohort, inclusion criteria were male and female volunteers aged 60 years or older, receiving influenza vaccine (Chiromas^®^ 2021/2020, adjuvant MF56C, Seqirus, Barcelona, Spain), and requiring hospitalization or ambulatory follow-up by palliative support teams (PADES). For the COVID-19 cohort, inclusion criteria were male and female volunteers aged 18 years or older, receiving COVID-19 vaccine (Comirnaty^®^, Pfizer-BioNTech), admitted to long-stay centers attached to Hospital Mare de Déu de la Mercè, or healthcare workers of the study centers. All participants were required to be able to take the study product orally and to understand the study procedures. Exclusion criteria were the need for assisted ventilation, any concurrent acute or chronic medical condition or medication that, in investigators’ opinion, may interfere with the study procedures or adherence to the study supplementation product (e.g., swallow problems, cognitive decline, short-stay hospital admission), hypersensitivity or allergy to the study product, terminal illness, and refusal to provide informed consent, as well as contraindication of the COVID-19 vaccine and clinical instability due to frailty or comorbid conditions.

An independent statistician generated a computer-based, sex- and age-balanced randomization list, and participants were randomly assigned in a 1:1 ratio to a dietary intervention with the nutritional supplement or supplementation with placebo. After randomization, participants were pooled into groups of 30–35 people, assigned to either the influenza or the COVID-19 cohorts according to fulfillment of the corresponding inclusion criteria.

### 2.3. Intervention and Study Procedures

Participants were administered the active product or placebo (maltodextrin) during 30 days in the influenza vaccine cohort and during 35 days in COVID-19 vaccine cohort. The active product (ABB C1^®^, AB Biotek Laboratories, a business division of AB Mauri, Peterborough, UK) was composed of a synergistic combination of yeast-based ingredients with a unique β-1,3/1,6-glucan complex and a consortium of heat-treated probiotic *S. cerevisiae* rich in selenium and zinc. A single-dose stick provides 516.67 mg of β-1,3/1,6-glucan, 233.33 mg of *S. cerevisiae* mineral yeast, 100 µg of selenium, and 15 mg of zinc. Participants were instructed to take a daily, single-dose stick of 3 g containing 750 mg of ABB C1^®^ or placebo orally with meals. The Pharmacy Service of Hospital Mare de Déu de la Mercè was responsible for supplying the food supplement in labelled stick sachets with identical appearances. Treatment assignment (ABB C1^®^ or placebo) was concealed until the end of the study.

Study procedures are shown in Figure 1. At the baseline visit (day 0), eligibility criteria were checked, informed consent was obtained, a peripheral blood sample was drawn, and the nutritional supplement was provided. Influenza vaccine or the first dose of COVID-19 vaccine was administered on the same day or on subsequent days (study day 1); a short supplementation period before vaccination was allowed. Follow-up visits were scheduled on days 7 and 30 in the influenza vaccine cohort, and on days 7, 21 (second dose of COVID-19 vaccine), and 35 in the COVID-19 cohort. In all follow-up visits, blood samples were obtained and safety assessments were performed. Participants in the influenza cohort were contacted by phone on day 60 to assess hospital readmission. Serious or unexpected adverse events were screened every visit and participants were instructed to use a diary card for recording side effects.

### 2.4. Laboratory Analyses

Blood immunity parameters included levels of CD+4, CD8+T, and CD3+T cells and specific IgM and IgG antibodies against influenza and SARS-CoV-2. Blood levels of T lymphocyte subsets were measured by flow cytometry (Cytomics FC500, Beckman Coulter Life Sciences, L’Hospitalet de Llobregat, Barcelona, Spain) and results were expressed as cells/µL. Specific IgM and IgG antibodies to SARS-CoV-2 spike and nucleocapsid proteins were measured by automated chemiluminescent immunoassay (Architect i 1000 SR, Abbott Diagnostics, Madrid, Spain) and expressed as AU/mL. Influenza A virus IgG antibodies against influenza A antigens A/Victoria/3/75 (ATCC VR-822) strain and Influenza B virus IgG antibodies against influenza B antigens, B/Hong Kong/5/72 (ATCC VR-791) strain, were determined by electro-chemiluminescence (Virclia^®^ IgG monotest, Vircell, S.L., Granada, Spain) and expressed as avidity index (AI). Serum levels of selenium and zinc were determined using inductively coupled plasma mass spectrometry (IC-PMS) (7900 Agilent Technologies, Barcelona, Spain) and results were expressed as µg/dL. All assays were performed according to the manufacturer’s instructions.

### 2.5. Statistical Analysis

Sample size calculation assumed an alpha risk of 0.1 and a beta risk of 0.2 in a bilateral contrast; 41 subjects were required to detect a difference equal to or greater than 0.3% units in the late response (day 30) to the influenza vaccine (IgG level) with respect to the placebo group. A loss to follow-up rate of 10% was estimated, resulting in a sample size of 45 vaccinated participants assigned to the ABB C1^®^ nutritional supplementation and 45 to placebo, with a total of 90 patients regardless of the type of vaccine administered.

The per-protocol (PP) dataset was analyzed, that is, all participants who were enrolled in the two cohorts and completed the follow-up visits, whereas the intention-to-treat (ITT) population was used for safety analysis. Categorical variables were expressed as frequencies and percentages, and continuous variables, as mean and 95% confidence interval (CI). Longitudinal mixed models for repeated measurements (MMRM) were used to analyze outcomes in different time points with Tukey’s pairwise comparisons post hoc test. Analyses included the fixed, categorical effects of treatment, visit, and treatment-by-visit interaction, declaring the subject as a random effect. An unstructured (co)variance structure was used. Statistical significance was set at *p* < 0.05. All analyses were conducted using R version 4.0.0 (https://intro2r.com/citing-r.html, accessed on 24 November 2021).

## 3. Results

It was planned to include 45 participants in each cohort, but, due to recruitment difficulties, 72 participants (27 males and 45 females) were finally included, 29 of whom (40.2%) were older than 70 years of age. Assessments were performed at baseline, at day 7 (intermediate evaluation), and at day 30 (final evaluation) in the influenza vaccine cohort. Assessments in the COVID-19 vaccine cohort were performed at baseline, at day 7, day 21, and day 35. Of the 72 participants, 67 were treated with ABB C1^®^ (*n* = 33) or placebo (*n* = 34). Protocol violation (no vaccination) occurred in the remaining five participants and they were excluded from the analysis. Vaccinated participants who completed the study included 32 supplemented with ABB C1^®^ (COVID-19, *n* = 18; influenza, *n* = 14) and 32 treated with placebo (COVID-19, *n* = 14; influenza, *n* = 14).

### 3.1. T Lymphocytes

The results of blood levels of T lymphocytes subsets are shown in Table 1 for the COVID-19 vaccine cohort and in Table 2 for the influenza vaccine cohort.

In relation to the primary outcome of the study, changes of blood levels of CD4+T lymphocytes showed an increased from a mean of 910.7 (95% CI 741.9 to 1079.4) cells/µL at baseline to 1000.2 (95% CI 831.4 to 1168.9) after the second dose of COVID-19 vaccine in the group treated with the ABB C1^®^ supplement, whereas, in the group treated with placebo, there was a decrease from 1055.1 (95% 867.0 to 1240.2) cells/µL at baseline to 929.8 (95% CI 743.1 to 1116.5) cells/µL (Figure 2). These differences were not statistically significant.

In relation to changes of CD3+T and CD8+T lymphocytes, a similar trend to the CD4+T cells was observed, with an increase at the end of the study as compared with the baseline in the group treated with the ABB C1^®^ supplement and a decrease in the group treated with placebo, although statistically significant differences were not observed.

In the influenza vaccine cohort, the time course of T lymphocytes during the study was quite similar to changes observed in the COVID-19 vaccine cohort, with increases in the CD4+T cell count at the end of the study as compared with baseline, both in the ABB C1^®^ supplement group and in the placebo group, although the magnitude of the increase was somewhat greater in the ABB C1^®^ group (Figure 3). At the final visits, blood levels of CD3+T and CD8+T subsets increased in the ABB C1^®^ supplement group and decreased in the placebo group as compared with baseline. These differences did not reach statistical significance.

### 3.2. Immunoglobulins

Serum levels of immunoglobulins in the COVID-19 vaccine cohort are shown in Table 3. In relation to IgG levels in the COVID-19 vaccine cohort, there were statistically significant increases in the ABB C1^®^ supplement group when values at 7, 21, and 35 days were compared with baseline (within-group differences, *p* < 0.001). In the placebo group, increases in serum IgG levels were also significant when values at day 35 were compared with values at days 21 and 35 and were compared with baseline (within-group differences, *p* < 0.001). However, between-group differences were not statistically significant (Figure 4). Serum levels of IgM increased in both study groups, with significant differences in the within-group comparison of values at day 35 versus values at baseline in the ABB C1^®^ supplement group (*p* < 0.001).

In the influenza vaccine cohort, changes of anti-influenza A and B IgG levels were similar in the ABB C1^®^ supplement and placebo groups at baseline and during the study (Table 4).

### 3.3. Selenium and Zinc

As shown in Figure 5, serum levels of selenium and zinc showed a higher increase in subjects treated with the food supplement ABB C1^®^ than in those receiving placebo, both in the COVID-19 vaccine and influenza vaccine cohorts. In the COVID-19 vaccine cohort, serum levels of selenium increased in the ABB C1^®^ supplementation group, whereas there was a significant decrease in the placebo group (Turkey’s pairwise comparison, *p* = 0.024). Serum zinc levels also increased in the active supplementation group and decreased in the placebo arm. In the influenza group, increases of serum levels of both selenium and zinc in the ABB C1^®^ supplementation group at the end of the study on day 35 as compared with baseline were statistically significant (Turkey’s pairwise comparison, *p* < 0.001). In the placebo group, serum levels of selenium remained almost unchanged, and serum levels of zinc showed a slight increase of a lower magnitude than increases observed in the ABB C1^®^ supplementation group (Table 5).

Statistically significant differences (Turkey’s pairwise comparison) in any of the study variables according to age and sex were not found.

### 3.4. Tolerability and Safety

No cases of hospital readmission were recorded. The active study product was well tolerated. Nausea, vomiting, and diarrhea occurred in six participants assigned to treatment with ABB C1^®^ and in four participants treated with placebo. All these episodes were uneventful. Two serious adverse events resulting in exitus were reported, one in the ABB C1^®^ group (a patient with intestinal subocclusion and multiorgan failure) and one in the placebo group (a polymedicated patient with bronchoaspiration and history of cognitive impairment and schizophrenia). These two cases of exitus were considered unrelated to the study treatment.

## 4. Discussion

In the present study, the performance of the influenza vaccine was shown to produce a protective effect against COVID-19, with a possible explanation being due to the trained immune process, since all participants in the COVID-19 group were previously vaccinated against influenza. Influenza vaccines can induce non-specific activation of innate immune cells [10,34]. This mechanism has been demonstrated in the protective effect of BCG vaccination of human malaria infection [35]. The potential benefit of influenza vaccination mitigating critical outcomes in patients with SARS-CoV-2 infection, such as sepsis, stroke, deep vein thrombosis, and emergency and ICU admissions, was analyzed in a retrospective analysis of two cohorts of 37,377 patients, having either received or not received influenza vaccination between 6 months and 2 weeks prior to COVID-19 infection [36]. Data were analyzed using a strict propensity score matching procedure adjusted by age, race, gender, ethnicity, hypertension, hyperlipidemia, diabetes, obesity, heart disease, chronic obstructive pulmonary disease, and smoking. At 30, 60, 90, and 120 days of SARS-CoV-2-positive diagnosis, patients in the cohort that received the influenza vaccine showed a significantly reduced risk for all outcomes. The authors of the study suggest that influenza vaccination may exert a potential protective effect that could benefit populations without ready access to COVID-19 vaccination [36]. In the present study, none of the patients in the COVID-19 group received a diagnosis of SARS-CoV-2 infection. Our findings suggest that the use of influenza vaccination may be a viable option to attenuate the effects of SARS-CoV-2. Even the patients who received the COVID-19 vaccine may have some benefit, since SARS-CoV-2 vaccination does not provide complete immunity. However, more studies are necessary to better clarify a possible protective role of the influenza vaccine against SARS-CoV-2 infection.

A key finding of the study was that supplementation with ABB C1^®^, a β-glucan complex and consortium of *S. cerevisiae* enriched with selenium and zinc, in participants receiving the COVID-19 mRNA vaccine of Pfizer-BioNTech resulted in T lymphocyte responses after a single dose. Humoral and cellular immune responses were higher in concentrations in the ABB C1^®^-supplemented group as compared to the placebo group; moreover, responses did not differ by age or sex. As recently reported, antibody markers strongly inversely correlated with COVID-19 risk and directly correlated with vaccine efficacy [37]. In addition to the protective effect of COVID-19 mRNA vaccine per se, efficacy in participants with or without previous SARS-CoV-2 infection was 94.6% (95% CI 89.9–97.3) [37].

The onset of the immune responses after the first dose was detected already on day 7 in the case of T lymphocytes, IgM, and IgG antibodies. A peak of antibody titers was seen as early as at 35 days, when the same dose of supplementation was administered. Our results also suggest that a booster dose of COVID-19 mRNA vaccine in association with food supplementation with ABB C1^®^ was more reactogenic that the first dose. IgG antibody titers increased, and a greater effect on CD4+T lymphocytes was observed after the booster dose. Interestingly, cellular and humoral responses were similar for the two types of vaccines tested, influenza and COVID-19, belonging to different technology platforms (adenovirus and mRNA, respectively), thus indicating a potential of ABB C1^®^ to supplement broadly immunization practices.

ABB C1^®^ was able to restore healthy levels of selenium and zinc in immunosenescent patients who are shown to be well below desirable blood levels at the beginning of the trial, providing a reliable source of absorbable micronutrients relevant to the immune function. The associations between the blood concentrations of selenium and zinc and those administered in the ABB C1^®^ supplement increased during its administration and maintained over time, which describes the good bioavailability of the product, showing statistically significant differences in the time course vs. placebo; selenium levels at baseline were below the threshold of a healthy state (15 μg/dL), and levels were restored to healthy values after ABB C1^®^ supplementation. On the other hand, zinc supplementation decreases the incidence of infection in the elderly and improves cytokine imbalance and oxidative stress markers [38,39]. The duration of the immune response to the COVID-19 (mRNA) vaccine is still under investigation. Therefore, the idea of supplementing patients with low trace metal levels with zinc to balance what has been called the cytokine storm caused by SARS-CoV-2 is appealing. Reassuringly, zinc supplementation for the common cold caused by rhinoviruses and coronaviruses has been shown to reduce the duration of illness and symptoms [40]. Of note, no statistically significant differences according to age and sex were found for any of the variables evaluated. This could be due to the heterogeneity of the sample, as patients were complex, multipathological, polymedicated elderly patients, which generated a great variability in the sample.

The duration of the immune protection of the COVID-19 vaccine was evaluated until reaching at least 35 days of follow-up, showing its effectiveness. However, it would be interesting to continue with the follow-up since the duration of protection for vaccines against COVID-19 has not yet been clearly defined. Additionally, the absence of clinical symptoms and hospital readmissions after vaccination associated with ABB C1^®^ intake could suggest several independent or linked explanations for this significant vaccine/supplementation association, including a possible protective effect of vaccination against disease acquisition, which has an impact on hospitalization and mortality in persons at higher risk of severe disease, decreasing it. Thus, it also reduces the burden of care on the health care system [41].

In relation to the primary endpoint of the study, influenza and COVID-19 vaccines induced an increase in CD4+T cells. Most immune responses associated with vaccination are controlled by specific T cells of a CD4+ helper phenotype, which mediate the generation of effector antibodies, cytotoxic T lymphocytes (CTLs), or the activation of innate immune effector cells. It was shown that some vaccines, such as COVID-19 mRNA vaccines, induce broad CD4+T cell responses [42]. Moreover, enhancement of antibody production (IgM and IgG) in the COVID-19 vaccine group was shown by a significantly faster and more intense release of antibodies.

Enhancement of phagocytosis may be suggested by an increase in CD8+T cells, also called cytotoxic T lymphocytes or killer T-cells. CD8+T lymphocytes are responsible for phagocytosis in antitumoral and antiviral immune responses [43]. In addition, CD8+ T lymphocytes are involved in the generation of long-lasting memory in the T cell response (trained immunity) [43,44]. On the other hand, an increase in CD3+ T cells was found. CD3+T cells are involved in activating both the cytotoxic T cell (CD8+ naive T cells) and T helper cells (CD4+ naive T cells). CD3+T lymphocyte responses are considered as biomarkers of efficacy of the immune response for their involvement in T cell development, activation, and function in reaction to external and endogenous stimulants [45,46]. We cannot provide a plausible explanation for the decreases in T-cell subsets in the placebo group. It was shown that the age of the subjects or the strain of influenza virus contained in the vaccine could be determinants of the T-cell response to vaccination [47], but the underlying mechanisms remain unclear and need to be elucidated in further studies.

The current work provides further data suggesting an increase in phagocytosis or pointing out that the ABB C1^®^ product may stimulate trained immunity (given the boost effect observed with the second COVID-19 vaccination in the in the ABB C1^®^ supplemented groups). The second dose of the Pfizer-BioNTech vaccination regimen was considerably more immunogenic in the ABB C1^®^-supplemented groups in terms of humoral and cellular immunity, both of which were linked to an innate immune response.

Our study presents some strengths and limitations. It is one of the first studies in which the immune response to two types of vaccines adding a nutritional supplement as an adjuvant therapy was evaluated in a population at high risk of complications in case of suffering from influenza infection or COVID-19 disease. A further strength of the study is that the administration of the nutritional supplement in both vaccine groups (influenza and COVID-19) showed an increase in the immune response after starting supplementation, which was measured by changes of immune biomarkers including T cells, specific antibodies against influenza and/or SARS-CoV-2, and blood levels of selenium and zinc. However, the present study has some limitations due to the variability of the sample, since the study population mainly included pluripathological and polymedicated elderly people, and a reduced number of participants, so that some changes in the vaccine groups as compared with baseline were not significant from a statistical point of view, but clinically relevant. Further studies are needed to assess the relevance of the clinical effects of the ABB C1^®^ supplement in different populations and centers based on a larger sample size, as well as the effects of this nutritional supplement on immune cells in the absence of vaccination. The present findings contribute to a first approach in the understanding of the benefits that nutritional supplementation and immunological changes may have for patients’ health.

Finally, β-1,3/1,6-glucans of yeast (as contained in ABB C1^®^) are one of the most widely studied molecules that have proven their potential in trained immunity. Some yeast and fungal β-glucans can exert trained immunity capacities both at a local level (mucosa) or systemically, reaching hematopoietic organs. Some genus of yeast inhabiting the mucosa (i.e., *Saccharomyces* spp.) in a symbiotic relationship with host cells may represent a source of ramified yeast β-glucans, allowing keeping the innate immune system in a good fitness status [9].

## 5. Conclusions

The administration of a nutritional supplement (ABB C1^®^) based on a combination of β-glucan and probiotic *S. cerevisiae* yeasts enriched with selenium and zinc in volunteers in association with influenza and COVID-19 mRNA vaccines appeared to be able to stimulate trained immunity as compared with placebo, which indicates that ABB C1^®^ provides a reliable source of absorbable micronutrients relevant to enhance the immune function.

## Figures and Tables

**Figure 1 nutrients-13-04347-f001:**
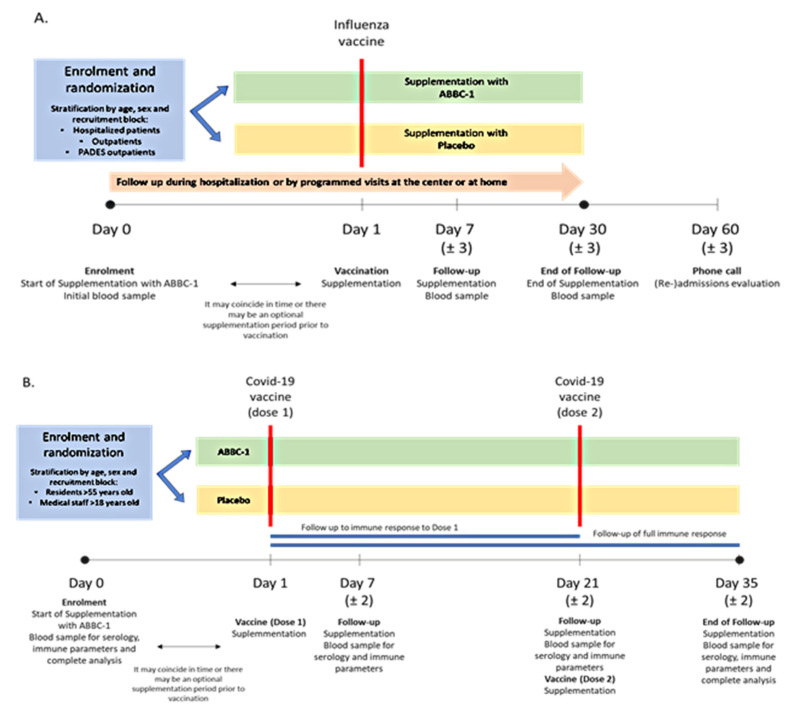
(**A**) In the influenza vaccine cohort, participants were given supplementation with ABB C1^®^ or placebo and blood samples for analyses were taken at baseline and at days 7 and 30 (end of study). (**B**) In the COVID-19 vaccine cohort, participants were given supplementation with ABB C1^®^ or placebo and received a first vaccine dose (day 1) and a second vaccine dose (day 21) and were followed for 35 days, with blood samples for analyses taken at baseline and at days 7, 21, and 35 (end of study).

**Figure 2 nutrients-13-04347-f002:**
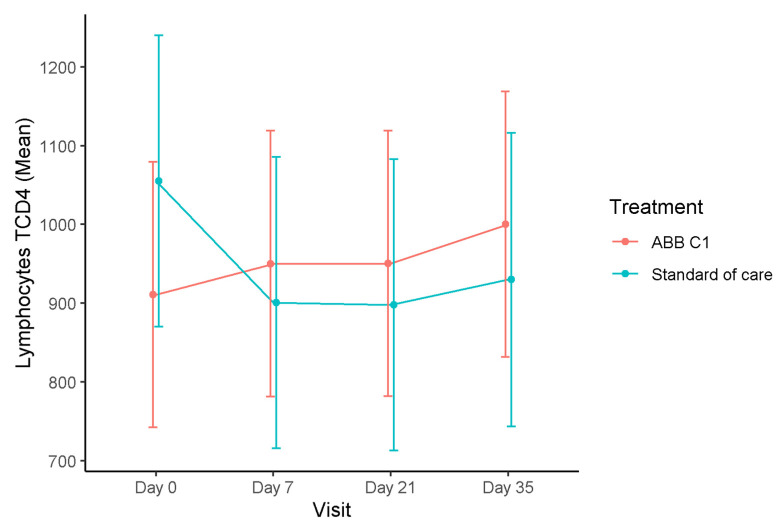
Time course of blood levels of CD4+T lymphocytes in the COVID-19 vaccine cohort. In the ABB C1^®^ supplementation group, CD4+T lymphocytes increased as compared with baseline and after the second vaccine dose, whereas, in the placebo group, there was a decrease as compared with baseline.

**Figure 3 nutrients-13-04347-f003:**
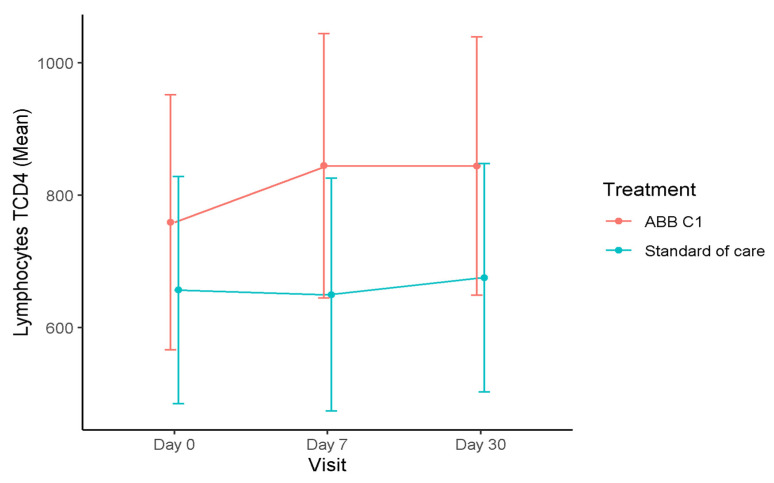
Time course of blood levels of CD4+T lymphocytes in the influenza vaccine cohort. There was an increase in CD+T lymphocytes at the end of the study (day 30) as compared with baseline in both supplemented groups, but the magnitude of the increase was greater in the ABB C1^®^ supplement group than in the placebo group.

**Figure 4 nutrients-13-04347-f004:**
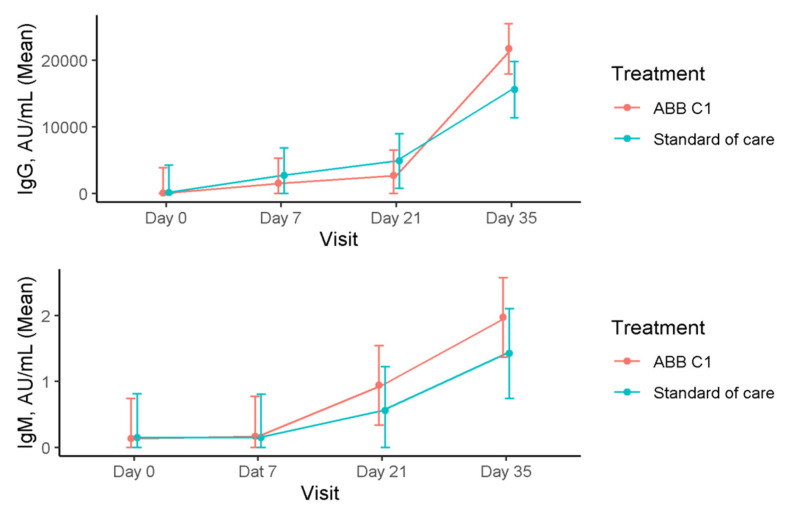
Time course of serum levels of IgG and IgM in the COVID-19 cohort during the study.

**Figure 5 nutrients-13-04347-f005:**
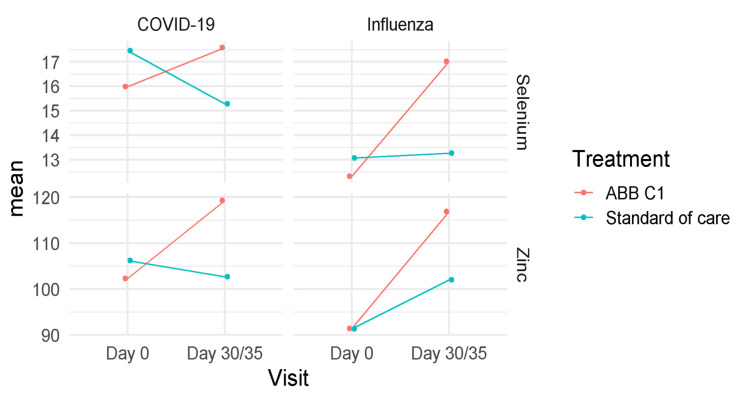
Changes of mean values of selenium and zinc concentrations in the COVID-19 and influenza vaccine cohorts at the end of the study as compared with baseline in the active supplementation (ABB C1^®^) and placebo (standard of care) groups.

**Table 1 nutrients-13-04347-t001:** Changes of CD4+T, CD3+T, and CD8+T cells during the study period in the COVID-19 vaccine cohort.

Study Cohort	CD4+T, Cells/µL Mean (95% CI)	CD3+T, Cells/µL Mean (95% CI)	CD8+T, Cells/µL Mean (95% CI)
COVID-19 vaccine			
ABB C1^®^ supplement			
Baseline (*n* = 18)	910.67 (741.90–1079.44)	1508.67 (1278.05–1739.28)	585.50 (464.27–706.73)
Day 7 (*n* = 18)	950.0 (781.23–1118.77)	1589.44 (1358.83–1820.06)	606.50 (485.27–727.73)
Day 21 (*n* = 18)	950.28 (781.51–1119.05)	1598.06 (1367.44–1828.67)	621.11 (499.88–742.34)
Day 35 (*n* = 18)	1000.17 (831.40–1168.94)	1699.56 (1468.94–1930.17)	653.44 (532.21–774.68)
Placebo			
Baseline (*n* = 15)	1055.07 (869.96–1240.18)	1587.47 (1334.52–1840.41)	496.73 (363.76–629.71)
Day 7 (*n* = 15)	900.60 (715.49–1085.71)	1367.20 (1114.25–1620.15)	435.73 (302.76–568.71)
Day 21 (*n* = 15)	897.80 (712.69–1082.91)	1391.67 (1138.72–1644.61)	450.87 (317.89–583.84)
Day 35 (*n* = 14)	929.80 (743.15–1116.46)	1407.23 (1151.75–1662.71)	450.10 (316.16–584.03)

CI: confidence interval.

**Table 2 nutrients-13-04347-t002:** Changes of T lymphocytes during the study period in the influenza vaccine cohort.

Study Cohort	CD4+T, Cells/µL Mean (95% CI)	CD3+T, Cells/µL Mean (95% CI)	CD8+T, Cells/µL Mean (95% CI)
Influenza vaccine			
ABB C1^®^ supplement			
Baseline (*n* = 15)	759.40 (566.85–951.45)	1195.20 (955.85–1434.55)	396.0 (291.55–500.45)
Day 7 (*n* = 12)	844.41 (644.80–1044.03)	1278.40 (1024.89–1531.91)	402.45 (291.22–513.69)
Day 30 (*n* = 14)	843.87 (648.94–1038.80)	1311.45 (1067.43–1555.46)	414.20 (307.54–520.87)
Placebo			
Baseline (*n* = 19)	656.84 (485.55–828.13)	1114.63 (901.71–1327.55)	422.68 (329.77–515.60)
Day 7 (*n* = 16)	649.93 (474.20–825.65)	1069.96 (847.99–1291.92)	390.68 (293.41–487.95)
Day 30 (*n* = 18)	675.42 (502.81–848.04)	1065.32 (849.68–1280.96)	359.61 (265.38–453.83)

CI: confidence interval.

**Table 3 nutrients-13-04347-t003:** Serum immunoglobulin levels in the COVID-19 cohort during the study period.

Study Cohort	IgG, AU/mL Mean (95% CI)	IgM, AU/mL Mean (95% CI)
COVID-19 vaccine		
ABB C1^®^ supplement		
Baseline (*n* = 18)	114.34 (−3634.77–3863.45)	0.14 (−0.46–0.74)
Day 7 (*n* = 18)	1560.97 (−2188.14–5310.08)	0.17 (−0.43–0.77)
Day 21 (*n* = 18)	2758.97 (−990.14–6508.08)	0.94 (0.34–1.54)
Day 35 (*n* = 18)	21,716.64 (17,967.53–25,465.75)	1.97 (1.37–2.58)
Placebo		
Baseline (*n* = 15)	136.95 (−3975.20–4249.10)	0.15 (−0.51–0.81)
Day 7 (*n* = 15)	2733.02 (−1379.13–6845.16)	0.15 (−0.51–0.81)
Day 21 (*n* = 15)	4892.81 (780.67–9004.96)	0.56 (−0.10–1.22)
Day 35 (*n* = 14)	15,595.70 (11,369.80–19,821.60)	1.42 (0.74–2.11)

CI: confidence interval.

**Table 4 nutrients-13-04347-t004:** Changes of anti-influenza A and B IgG antibodies during the study.

Study Cohort	Anti-Influenza A IgG, AI Mean (95% CI)	Anti-Influenza B IgG, AI Mean (95% CI)
Influenza vaccine		
ABB C1^®^ supplement		
Baseline (*n* = 15)	2.29 (1.08–3.50)	2.35 (0.92–3.78)
Day 7 (*n* = 12)	2.41 (1.19–3.63)	1.95 (0.48–3.42)
Day 30 (*n* = 14)	2.22 (1.01–3.43)	2.31 (0.86–3.75)
Placebo		
Baseline (*n* = 19)	2.67 (1.60–3.75)	2.54 (1.27–3.81)
Day 7 (*n* = 16)	2.37 (1.28–3.45)	2.42 (1.12–3.71)
Day 30 (*n* = 18)	2.26 (1.18–3.33)	2.42 (1.14–3.70)

AI: avidity index; CI: confidence interval.

**Table 5 nutrients-13-04347-t005:** Changes of serum levels of selenium and zinc in the study groups among participants in the COVID-19 vaccine and influenza vaccine cohorts.

Study Cohort	Selenium, µg/dL Mean (95% CI)	Zinc, µg/dL Mean (95% CI)
COVID-19 vaccine		
ABB C1^®^ supplement		
Baseline (*n* = 18)	15.98 (13.99–17.98)	102.33 (91.79–112.88)
Day 35 (*n* = 18)	17.57 (15.57–19.57)	119.28 (108.71–129.85)
Placebo		
Baseline (*n* = 15)	17.45 (15.26–19.64)	106.20 (94.63–117.77)
Day 35 (*n* = 14)	14.90 (12.67–17.13)	102.64 (90.66–114.63)
Influenza vaccine		
ABB C1^®^ supplement		
Baseline (*n* = 15)	12.33 (11.15–13.51)	91.40 (79.07–103–73)
Day 30 (*n* = 14)	16.94 (15.74–18.14)	116.75 (104.04–139.47)
Placebo		
Baseline (*n* = 19)	13.07 (12.02–14.12)	91.32 (80.35–102.28)
Day 30 (*n* = 18)	13.41 (12.35–14.47)	102.23 (91.01–113.45)

CI: confidence interval.

## Data Availability

Data of the study are available from the corresponding author (J.A.M.R.) upon request.

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
