# Peer review of "Effect and Tolerability of a Nutritional Supplement Based on a Synergistic Combination of β-Glucans and Selenium- and Zinc-Enriched Saccharomyces cerevisiae (ABB C1®) in Volunteers Receiving the Influenza or the COVID-19 Vaccine: A Randomized, Double-Blind, Placebo-Controlled Study"

_nutrients, 2021, doi:10.3390/nu13124347_

Round 1

Reviewer 1 Report

The study is interesting and very appropriate for the journal. The authors’ English is excellent.

There are a few comments. There is an (apparent) discrepancy between the figures on lines 176 (90) and 187 (72).  Does ‘included’ rather than ‘consisted of’ mean that others were involved?

On page 6 and 7, lines 204 and 225 can the authors explain the decrease in CD4+T, CD3+T and CD8+T in placebo group? And was the increase of these subsets significant in ABB C1 group?

Author Response

Please see the attachment. the section

AUTHORS’ COMMENTS TO SUGGESTIONS OF REVIEWER #1

kinds regards

Reviewer 2 Report

The authors Mateus et al present “Efficacy and tolerability of a nutritional supplement based on a synergistic combination of β-glucans and selenium- and zinc-enriched Saccharomyces cerevisiae (ABB C1®) in volunteers receiving the influenza or the COVID-19 vaccine: a randomized, double-blind, placebo-controlled study”, a study detailing a human trail aimed at interrogating any effects of Selenium and zinc supplementation on immune responses generated through vaccination.

The topic is interesting and worthy of investigation and publication. However, in my opinion major corrections and restructuring are required to increase the quality of the manuscript – and these corrections surround the way the authors have presented the data, rather than requiring extensive new experimentation.  With major corrections to the way the authors have presented the data, and discussed it, the manuscript can be improved considerably – improving how useful it is to the reader, and subsequently how successful the paper is.

I have detailed major “general” comments, and line by line comments for the authors to consider.

Major:

  1. The effect of the treatment on immune cells in the absence of vaccination is not properly evaluated. No mention of the properties of the vaccines is mentioned (e.g any adjuvant in the influenza vaccine) which could alter the results of the study
  2. This is not an efficacy study, the authors have not shown any efficacy data, and the pool of patients is likely too low to make this work as an efficacy study.
  3. Results and discussion backing up the claim that an influenza vaccine increases protection against COVID-19 is not sufficient in this manuscript
  4. Statistical analysis is weak in this manuscript, and the authors take too much confidence from results presented without statistics to back up whether the differences are significant or not.
  5. The specificity of T cells and Ig has not been detailed sufficiently in this manuscript, which protein or peptide target? Etc. Which assays were used to show immunogenicity of the vaccines? Or are all results non-specific?
  6. The discussion is too strongly worded based on the results. For example, an increase in CD8+ T cells is not a suitable secondary measure to state “enhancement of phagocytosis”, the downstream assays should be completed to state this. The study does not validate the use of ABB C1 stimulating trained immunity, and this is still poorly understood. “Clinical evidence” that influenza vaccination protects against severe COVID-19 is not present in this manuscript, and I also disagree that “molecular evidence” was presented.

Line by line responses:

Line 47, the influenza vaccine does not protect against sars-cov-2, this is misleading, the vaccine itself will not protect against SARS-CoV-2, but may activate the immune system sufficiently to prevent infection (by any virus) in a window after vaccination.

Line 61 to 63, citation for comments on the properties of selenium

Line 75, “protecting epithelial cells from the attachment of pathogens” Can the authors clarify what they are claiming here? That these supplements are preventing viral attachment?

Line 95, relevance of CD4+T cells to  the general term “immunity” used in this manuscript, it seems that only T cell counts and overall IgG and IgM are reported, ideally specific tests can be performed against influenza and SARS-CoV-2 viruses, these assays are widely available.

Line 98, “serum levels”? Do the authors mean cell counts found from blood?

Line 117 “any medical condition that in investigators’ opinion may interfere”  Please define the process

Line 128 “in during 30 and 35 days” this is not clear, when the substance was administered, please clarify.

Line 131, “composed of”

Line 131 “synergistic” – is there a citation for this claim of synergy, in the context of this study?

Line 133, do the authors have the doses of selenium and zinc within the mineral yeast? Can the authors confirm that “a single-dose stick” is 3g as suggested in line 135?

Line 147 “serious or”

Line 155, figure 1 legend needs to be expanded to include all the details in the figure (Stand-alone)

Line 159, plasma containing cells? Or cells purified prior to the processing of blood into plasma?

Line 160, antibodies to viruses, not to diseases (I.e. SARS-CoV-2 spike, not to COVID-19)

Line 185, citation for R would be beneficial https://intro2r.com/citing-r.html

Line 187, “72 volunteers” but “Twenty-nine participants”? why the change in style? Suggestion to authors to change to “29 participants” to retain the same style

Line 196, expand Table  1 legend to include all info to evaluate the table data.

Line 210, expand figure 2 legend

Any statistics on the significance of the differences observed between vaccine groups? Statistics on figure 3 too.

Line 231, figure 3 legend updating

Hard to associate any benefit of ABB C1

Line 246, figure 4 is misleading with only start and end points plotted, and very poor quality, improve or remove. Expand legend if improving

Table 4, there seems to be no effect of the influenza vaccine on influenza virus titres

Which antigens were flu and covid tests done on?

Line 263, which test was carried out?

Line 285, it may be beneficial to the reader to state which tests were carried out

Line 297 “protective effect against COVID-19” how

Line 360, “enhancement of phagocytotis” was not measured by the authors, this must be rephrased

Line 370, disagree that anything has been validated in this study, as there is a lack of statistics backing this up.

Author Response

Please see the attachment. Section

AUTHORS’ COMMENTS TO SUGGESTIONS OF REVIEWER #2

Kinds regards

Round 2

Reviewer 2 Report

The authors have made substantial corrections to the manuscript and it is now suitable for publication.